# Characteristics of Clinical Trial Participants with Duchenne Muscular Dystrophy: Data from the Muscular Dystrophy Surveillance, Tracking, and Research Network (MD STAR*net*)

**DOI:** 10.3390/children8100835

**Published:** 2021-09-23

**Authors:** Katherine D. Mathews, Kristin M. Conway, Amber M. Gedlinske, Nicholas Johnson, Natalie Street, Russell J. Butterfield, Man Hung, Emma Ciafaloni, Paul A. Romitti

**Affiliations:** 1Carver College of Medicine, The University of Iowa, Iowa City, IA 52242, USA; katherine-mathews@uiowa.edu; 2Department of Epidemiology, The University of Iowa, Iowa City, IA 52242, USA; kristin-caspers@uiowa.edu; 3Department of Internal Medicine, The University of Iowa, Iowa City, IA 52242, USA; amber-gedlinske@uiowa.edu; 4Department of Neurology, Virginia Commonwealth University, Richmond, VA 23298, USA; nicholas.johnson@vcuhealth.org; 5Centers for Disease Control and Prevention, National Center on Birth Defects and Developmental Disabilities, Atlanta, GA 30329, USA; ntl2@cdc.gov; 6Departments of Pediatrics and Neurology, University of Utah, Salt Lake City, UT 84132, USA; russell.butterfield@hsc.utah.edu; 7College of Dental Medicine, Roseman University of Health Sciences, South Jordan, UT 84095, USA; mhung@roseman.edu; 8Department of Neurology, University of Rochester, Rochester, NY 14642, USA; Emma_Ciafaloni@URMC.Rochester.edu

**Keywords:** clinical trials, Duchenne muscular dystrophy, public health surveillance

## Abstract

Background: Therapeutic trials are critical to improving outcomes for individuals diagnosed with Duchenne muscular dystrophy (DMD). Understanding predictors of clinical trial participation could maximize enrollment. Methods: Data from six sites (Colorado, Iowa, Piedmont region North Carolina, South Carolina, Utah, and western New York) of the Muscular Dystrophy Surveillance, Tracking, and Research Network (MD STAR*net*) were analyzed. Clinical trial participation and individual-level clinical and sociodemographic characteristics were obtained from medical records for the 2000–2015 calendar years. County-level characteristics were determined from linkage of the most recent county of residence identified from medical records and publicly available federal datasets. Fisher’s exact and Wilcoxon two-sample tests were used with statistical significance set at one-sided *p*-value (<0.05) based on the hypothesis that nonparticipants had fewer resources. Results: Clinical trial participation was identified among 17.9% (MD STAR*net* site: 3.7–27.3%) of 358 individuals with DMD. Corticosteroids, tadalafil, and ataluren (PTC124) were the most common trial medications recorded. Fewer non-Hispanic blacks or Hispanics than non-Hispanic whites participated in clinical trials. Trial participants tended to reside in counties with lower percentages of non-Hispanic blacks. **Conclusion**: Understanding characteristics associated with clinical trial participation is critical for identifying participation barriers and generalizability of trial results. MD STAR*net* is uniquely able to track clinical trial participation through surveillance and describe patterns of participation.

## 1. Introduction

The dystrophinopathies, Duchenne muscular dystrophy (DMD) and allelic Becker muscular dystrophy (BMD), are X-linked recessive disorders caused by mutations in the *DMD* gene that result in deficient dystrophin production [1]. Dystrophin is a cytoplasmic protein expressed in skeletal and cardiac muscle that links the contractile matrix to the sarcolemmal membrane, providing structural integrity during contraction. In general, DMD is associated with mutations that disrupt the reading frame or lead to a premature stop, resulting in a complete absence of dystrophin in muscle. DMD is characterized by progressive muscle weakness. Historically, males with DMD lost independent ambulation by 12 years of age and had a significantly shortened life span with death in the second or third decade of life from respiratory failure or cardiomyopathy [1,2,3]. Although DMD has a relatively stereotyped progression, variability is observed in the age at onset of muscle weakness and rate of progression [4]. BMD is associated with reduced or abnormal dystrophin protein and mutations that retain reading frame. BMD includes a wide spectrum of severity and is historically distinguished from DMD by having loss of independent ambulation after age 16 years.

Although advances in the multidisciplinary care, especially regarding respiratory care and use of corticosteroids, have produced considerable improvement in life expectancy [3,5,6,7,8], therapeutic trials are critical for improving outcomes in males with Duchenne muscular dystrophy (DMD) or Becker muscular dystrophy (BMD). ClinicalTrials.gov currently lists 39 interventional studies for males with DMD or BMD that are pending or actively enrolling children from birth to 17 years of age (date accessed 5 November 2021). Of these interventional studies, 24 involve some form of therapeutic drug intervention. To complete therapeutic trials as quickly as possible and have the statistical power to determine effectiveness of the therapies under study, it is desirable for all males who are eligible for therapeutic trials to have the opportunity to participate [9].

Participation in therapeutic clinical trials is often taxing on families, typically requiring missed days from work and school, travel, and long days of evaluation; thus, participation may be skewed towards those with more resources [10,11,12,13]. Further, the willingness to participate in or have access to clinical trials varies by race/ethnicity, socioeconomic status, and geographic location [14,15,16,17,18]. The Muscular Dystrophy Surveillance, Tracking, and Research Network (MD STAR*net*) ascertains individuals diagnosed with DMD or BMD and collects demographic and longitudinal clinical data, including clinical trial participation. Our study describes baseline associations between selected clinical and sociodemographic characteristics and participation in a therapeutic clinical trial for individuals diagnosed with Duchenne muscular dystrophy (DMD) followed by the MD STAR*net* during 2000–2015.

## 2. Materials and Methods

MD STAR*net* is a population-based public health surveillance program of nine muscular dystrophies, including dystrophinopathies, funded by the Centers for Disease Control and Prevention. Details about MD STAR*net* were published previously [19,20,21,22,23,24]. For this study, retrospective active surveillance data were collected by six sites: Colorado (CO), Iowa (IA), North Carolina, Piedmont region (NC), South Carolina (SC), Utah (UT), and New York’s 21 western counties (wNY). Eligibility for case inclusion in this study included having data on dates of birth and diagnosis of DMD or BMD on or after 1 January 2000, residency in an MD STAR*net* surveillance region, and a confirmed health encounter during 1 January 2000–31 December 2015.

Retrospective medical record review was completed for encounters during the eligibility period using a standardized abstraction tool that documented signs and symptoms (trouble rising/Gowers’ sign, trouble walking/running/jumping, frequent falling/clumsiness, inability to keep up with peers, abnormal gait, loss of motor skills, gross motor delay, or muscle weakness), ambulation status, family history, results of diagnostic testing (genetic test; muscle biopsy—immunostaining, Western blot; CK), health encounters, medical test results documenting functioning within the respiratory, cardiac and skeletal systems, medical interventions for each system, and medications prescribed.

Using abstracted signs and symptoms, clinical test results, and family histories, individuals were assigned a clinical classification by clinical review and consensus (see Figure 1) [19]. Data for all individuals, excluding those classified with possible DMD or BMD, were pooled to create an analytical dataset. We restricted our analyses to individuals classified as definite DMD (*n* = 371) as this group was genetically confirmed. We also excluded individuals ascertained by UT, but who were residents of Nevada, due to incomplete medical record access (*n* = 13). Our final sample was comprised of 358 individuals with DMD.

### 2.1. Clinical Trial Participation

We defined clinical trial participation based on the evidence of receiving a therapeutic non-steroidal clinical trial medication (type, year); enrollment in a corticosteroid clinical trial (year); or a checkbox for any clinical trial participation (yes/no). These indicators were combined to determine any clinical trial participation. Unless otherwise indicated, we included all individuals with any evidence of clinical trial participation. Because age at clinical trial participation was not available across all indicators of participation, we estimated the age of the individual in 2015 to describe the age distribution at the end of the surveillance period.

### 2.2. Clinical Characteristics

Using month and year, we estimated the age at loss of independent ambulation and classified ambulation status as walking, not walking, or unknown status. Family history of Duchenne or Becker MD was categorized as definite, suspected, no known family history, or unknown family history. Mutation type was classified as deletion, duplication, point mutation and double mutation, or unknown. Using complete dates, we calculated the age at first and last clinic visits. Finally, corticosteroid use was classified as no or yes.

### 2.3. Sociodemographic Characteristics

We collected race/ethnicity from the medical record for the child and from the birth certificate for the child and parents, where available. Because MD STAR*net* did not collect parent education and race/ethnicity from the medical record and there were limitations for linking to birth certificates for all individuals (missing ranged from 10.9–52.3% across sites), we linked the most recent county residence information collected from the medical record to publicly available national datasets that provide county-level population data for race/ethnicity, education, and economic indicators of poverty and household income. For race/ethnicity, we used population estimates for males aged 5 to 9 years in 2015 and estimated the percent of the population that was non-Hispanic white alone, non-Hispanic black alone, non-Hispanic other or combined races, and Hispanic. We used the 5-year average from the 2014–2018 American Community Survey for education and analyzed the percentage of the population that had less than a high school education, high school degree, some college, and a bachelor’s degree. The 2015 economic indicators represented the median percentage of the population that met the poverty guideline and the median household income. Finally, the 2013 Rural-Urban-Continuum Codes were recategorized into the metropolitan area >1,000,000, metropolitan area 250,000–1,000,000, metropolitan area <250,000, or nonmetropolitan urban or rural adjacent to metropolitan area, and nonmetropolitan urban or rural not adjacent to metropolitan area.

### 2.4. Statistical Analysis

We compared percentages and mean values by clinical trial participation. Fisher’s exact tests were used to test associations between categorical variables and Wilcoxon two-sample tests were used for continuous variables. Based on the hypothesis that clinical trial non-participants had fewer resources, the one-sided *p*-value of <0.05 was used to determine statistical significance.

## 3. Results

Therapeutic clinical trials for DMD active during 2000–2015 are presented in Figure 2. Overall, 17.9% of 358 individuals ascertained by the MD STAR*net* were identified as participating in a clinical trial (Table 1). During the period of this study, there were mutation-specific clinical trials open for patients with nonsense mutations (premature stop codons) and deletions that would have restored the reading frame by skipping exon 51.

Of all individuals, 9.5% had a mutation amenable to exon skipping treatments and 10.3% had nonsense mutations (Table 1). Of those eligible to receive a non-corticosteroid clinical trial medication, no single medication was taken by more than 15 individuals. Corticosteroids, tadalafil, and ataluren (PTC124) were the most common clinical trial medications recorded. The age of those individuals in 2015 who had participated in a clinical trial during 2000–2015 tended to cluster between 7 and 12 years (Figure 3) and most clinical trial medications were identified as having started during 2013–2015 (Figure 4).

Clinical trial participation differed by MD STAR*net* site, ranging from 3.7 to 27% of individuals between sites (Table 2). Individuals were identified as receiving a clinical trial medication in five sites, but participation in a corticosteroid clinical trial was only identified in two (data not shown). Clinical trial participants were more likely to be non-Hispanic white; fewer non-Hispanic blacks and Hispanics were identified as participants (Table 2). For county-level sociodemographic factors, individuals who participated in a clinical trial resided in counties with lower percentages of non-Hispanic blacks (Table 2). No statistically significant differences by clinical trial participation were found for county-level education, household income, household poverty, nor rural-urban continuum codes.

Among the clinical characteristics examined, individuals who were still walking were more likely to have enrolled in a trial and most had documented corticosteroid use unrelated to participation in a clinical trial. (Table 3). No differences by clinical trial participation were found for ages at first and last clinic visits nor family history of DMD. Although our methods do not allow us to identify mutation specific trials, 14 of 37 (37.8%) individuals with a nonsense mutation and 5 of 34 (14.7%) individuals amenable to exon skipping participated in a clinical trial during this period (data not shown). 

## 4. Discussion

DMD clinical trial participation increased from 2013 through 2015, consistent with the increasing number of phase 2 or 3 clinical trials listed on ClinicalTrials.gov. Overall, we report nearly 18% of individuals identified by the MD STAR*net* had participated in a clinical trial, which is consistent with enrollment reported by a 2013–2016 Muscular Dystrophy Association highlight of the findings from their neuromuscular disease registry [25]. We identified variation in participation by MD STAR*net* site and by race and ethnicity, at the individual or county level.

The differences we observed across MD STAR*net* site were largely due to two sites’ proximity to corticosteroid clinical trials and less than expected participation at one site that does not have an MDA clinic within their surveillance region. These observations suggest the importance of family proximity to a clinical trial site, resulting in the lower burden of travel and time away from home, and emphasize the need for geographically dispersed trial sites. Our observations are also consistent with parental reports of the clinic as an important source of information about clinical trials in general and specific clinical trial options [13]. Because clinics that care for individuals with DMD differ in resources and knowledge, there is the need for broad dissemination of information about clinical trials outside of the clinic setting, a role that patient advocacy groups have increasingly taken on [10,26].

Participation in a clinical trial is not only dependent on knowledge about the trial and willingness to enroll, but also requires meeting the inclusion/exclusion criteria for the trial of interest. Each trial has distinct entry criteria, but ambulatory males aged 7–10 years who are taking corticosteroids were historically the most common target population. This group is generally able to engage in motor function outcome measures, they have a predictable rate of change without treatment, and they are at a stage of disease when meaningful change can be detected [10]. We did not find age to be a significant factor in trial participation, but our cohort was generally 5–10 years old during the period of observation. Depending on the mechanism of action of the therapeutic agent being investigated, there might be additional limitations in participation for patients with specific mutations or types of mutations. Our description of the mutations among the clinical trial participants are consistent with the targeted clinical trials open during that time.

Of the individual characteristics examined, race/ethnicity was significantly associated with participation. Compared to non-Hispanic whites, lower participation was found among non-Hispanic blacks and Hispanics. The county-level results were partly consistent; those counties in which there was lower participation had higher percentages of non-Hispanic blacks. The observation that non-Hispanic blacks and Hispanics had a lower participation rate than non-Hispanic whites is consistent with observations in other diseases [27]. We note that although it was observed that non-Hispanic blacks have a lower prevalence of DMD than non-Hispanic whites [21,24], in this study we determined the rate of participation among identified patients, so the findings cannot be explained by this diagnostic discrepancy. Many explanations were proposed for discrepancies in trial participation by race or ethnicity [26,27]. One factor that influences the probability of a parent agreeing to enroll their child in a clinical trial is trust in the medical researchers. In a study of factors influencing adolescent’s parents’ trust regarding clinical trials, race, education level, and clinical trial impact on the child were the most significant predictors of parental trust [28]. In our population-based study, education level as estimated by county of residence was not associated with the likelihood of trial participation. It is of interest that of all US trials undergoing an FDA review between 1995 and 1999 (primarily adult trials), for which data on race of participants were available, the percentage of non-Hispanic black participants was nationally representative and Hispanic and other non-white racial groups were under-represented [29]. Our data add to the knowledge base that highlights the importance of efforts to ensure clinical trial participation that fully represents the affected population. Consistent and accurate information available to families and communities is one approach supported by previous research [27]. This can occur in the clinical setting with a trusted provider offering information and through the efforts of advocacy organizations.

### Strengths and Limitations

The MD STAR*net* conducts population-based surveillance and includes sites throughout the United States. The systematic collection of clinical and sociodemographic data from all eligible individuals, with DMD, regardless of clinical trial participation status, allows a comprehensive characterization of both participants and nonparticipants. Further, clinical data are reviewed by a team of specialists and diagnoses are systematically confirmed. Limitations include the reliance on medical records to identify clinical trial participation. Information about clinical trial participation is largely collected outside of primary medical records and may only be identified if noted by the provider who is managing neuromuscular care of the individual. Further, data completeness may differ by the type of medical record source (tertiary care, independent clinic). MD STAR*net* retrospectively identifies and longitudinally follows eligible individuals. However, loss to follow up due to a movement out of a surveillance site or receipt of care at a source not accessible by the surveillance program may underestimate the number of trial participants.

## 5. Conclusions

MD STAR*net* is uniquely positioned to identify and describe those who are not clinical trial participants. Observing differences between clinical trial participants and non-participants is critical in helping us understand possible barriers to participation and maximize generalizability of trial results. MD STAR*net* is also able to provide population-based information that describes the proportion of the patients within the surveillance areas who are participating in trials, i.e., the degree of saturation of the target population. Our established methods for monitoring clinical trial participation through surveillance and identifying a cohort for prospectively tracking patterns of clinical trial participation provides a unique opportunity to track success of the FDA initiative to reduce disparities in clinical trial research [30].

## Figures and Tables

**Figure 1 children-08-00835-f001:**
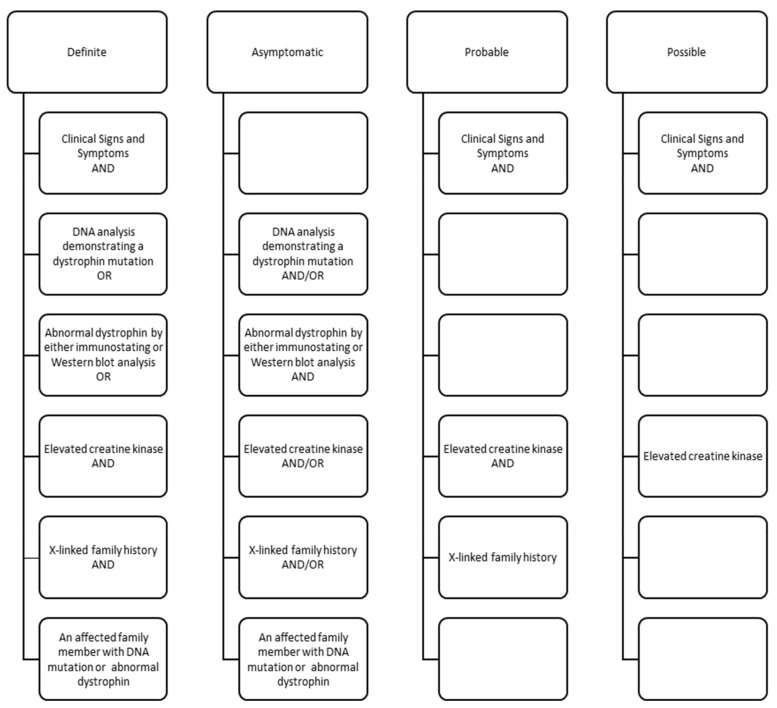
Clinical components for classification of males with Duchenne or Becker muscular dystrophy in the Muscular Dystrophy Surveillance, Tracking and Research Network (MD STAR*net*), 2000–2015. Classifications defined as: definite–clinical signs plus direct confirmation (pathogenic *DMD* mutation, decreased amount or size of dystrophin, or X-linked family history and elevated creatine kinase); asymptomatic–pathogenic *DMD* mutation but no clinical signs; probable–clinical signs plus elevated creatine kinase and X-linked family history; possible–clinical signs plus elevated creatine kinase, but no family history.

**Figure 2 children-08-00835-f002:**
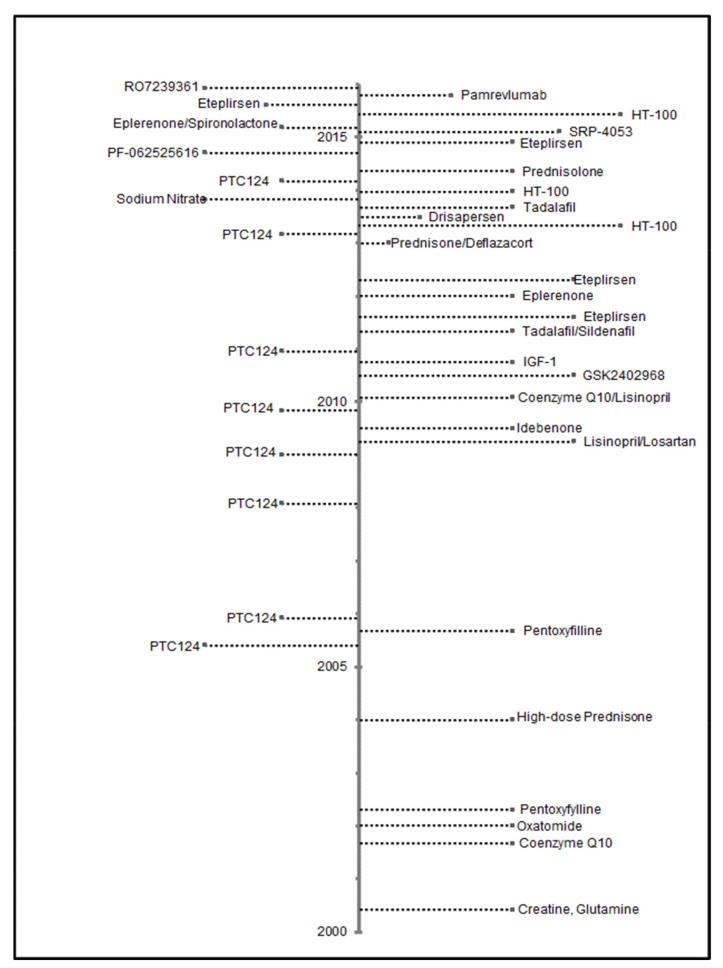
Dates of posting Phase 2 and 3 drug trials from ClinicalTrials.gov, using search criteria of Duchenne muscular dystrophy as disease; interventional studies as study type; and child (birth to 17 years) age. Trials were additionally limited to those occurring in the U.S. with a start date from 1 January 2000 to 31 December 2015.

**Figure 3 children-08-00835-f003:**
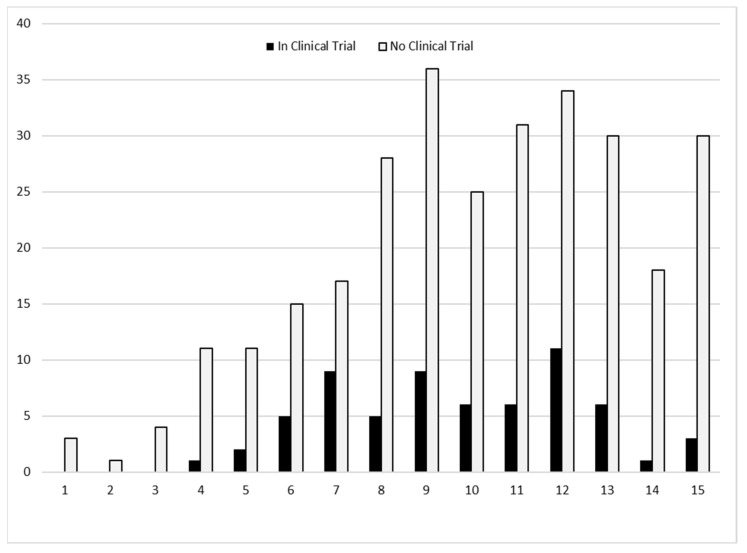
Number of individuals by age (years) in 2015 and clinical trial participation during 2000–2015, the Muscular Dystrophy Surveillance, Tracking, and Research Network (MD STAR*net*).

**Figure 4 children-08-00835-f004:**
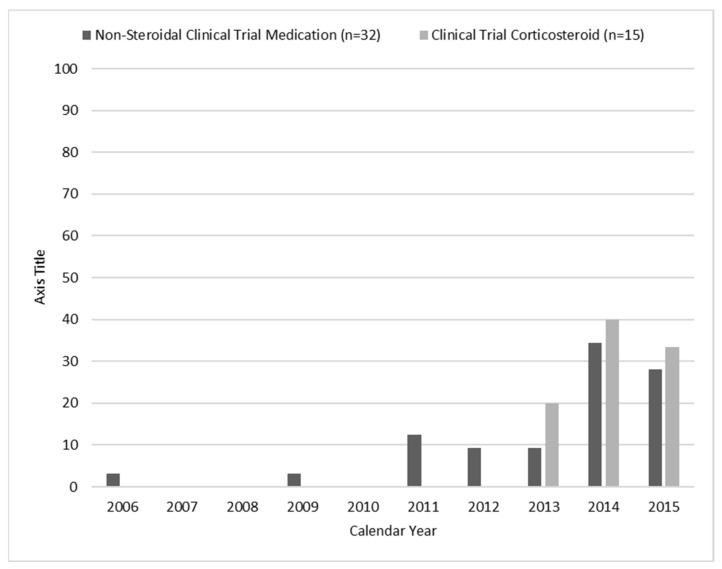
Frequency percentages for calendar year of initial documentation of enrollment in a clinical trial, the Muscular Dystrophy Surveillance, Tracking, and Research Network (MD STAR*net*), 2000–2015. Note: Year of use was missing for non-steroidal clinical trial medication (*n* = 1).

**Table 1 children-08-00835-t001:** Characteristics of Clinical Trial Eligibility and Participants in the Muscular Dystrophy Surveillance, Tracking, and Research Network (MD STAR*net*), 2000–2015.

Characteristics	*n* (%)
Total Sample	358
Clinical Trial Eligible Mutations, All Individuals	
Exon-Skippable (exon 51) Deletions, *n* (% total)	34 (9.5)
Nonsense (premature stop codon) Mutations, *n* (% total)	37 (10.3)
Clinical Trial Participation, *n* (% total) ^1^	64 (17.9)
Clinical Trial Checkbox	59 (16.5)
Corticosteroid Clinical Trial	15 (4.2)
Clinical Trial Medication (non-corticosteroid)	33 (9.2)
Clinical Trial Medication, *n* (% total medications) ^2^	33
Tadalafil	9 (27.3)
Ataluren (PTC124)	9 (27.3)
Idebenone	4 (12.1)
Drisapersen (GSK2402968)	3 (9.1)
CAT-1004	2 (6.1)
IGF-1	2 (6.1)
Domegrozumab (PF-06252616)	1 (3.0)
Eplerenone	1 (3.0)
Eteplirsen	1 (3.0)
Unspecified	1 (3.0)

^1^ More than one clinical trial category may be identified per individual. ^2^ More than one clinical trial medication may be identified per individual.

**Table 2 children-08-00835-t002:** Sample characteristics by any clinical trial participation in the Muscular Dystrophy Surveillance, Tracking, and Research Network (MD STAR*net*), 2000–2015.

Characteristics	In Clinical Trial	Not in Clinical Trial	*p*-Value
Total	64	294	
MD STARnet Site, *n* (row %)			0.0118 ^1^
Site A	15 (17.0)	73 (83.0)	
Site B	10 (17.9)	46 (82.1)	
Site C	10 (17.0)	49 (83.0)	
Site D	12 (26.0)	34 (74.0)	
Site E	2 (3.7)	52 (96.3)	
Site F	15 (27.3)	40 (72.7)	
Child characteristics			
Child Race/Ethnicity, *n* (row %)			0.0486 ^1^
non-Hispanic white	52 (21.4)	191 (78.6)	
non-Hispanic black	1 (4.0)	24 (96.0)	
non-Hispanic other	2 (15.4)	11 (84.6)	
Hispanic or Latino/Latina	7 (10.8)	58 (89.2)	
Unknown	2	10	
County-level sociodemographics			
Race/ethnicity (2015), Mean (95% CL) ^3^			
non-Hispanic white alone	65.5 (61.1, 70.0)	62.4 (60.6, 64.3)	0.1019 ^2^
non-Hispanic black alone	8.1 (5.5, 10.7)	11.6 (10.1, 13.1)	0.0115 ^2^
non-Hispanic other or combined	7.4 (6.6, 8.1)	7.4 (7.1, 7.7)	0.4596 ^2^
Hispanic	19.0 (16.4, 21.7)	18.6 (17.2, 20.0)	0.1383 ^2^
Education (2014–2018), Mean (95% CL) ^4^			
Less than HS	9.6 (8.9, 10.2)	10.2 (9.8, 10.7)	0.3228 ^2^
HS	26.1 (24.4, 27.9)	26.3 (25.5, 27.0)	0.4385 ^2^
Some college	31.7 (30.7, 32.7)	31.5 (31.0, 31.9)	0.3416 ^2^
Bachelor’s degree	32.6 (30.1, 35.2)	32.0 (30.8, 33.2)	0.4536 ^2^
Economic indicators ^5^			
Household income (2015), median (95% CL)	$58,367 ($56,067, $60,667)	$57,148 ($55,920, $58,376)	0.1631 ^2^
Poverty (2015), median percent of population (95% CL)	17.0 (15.6, 18.4)	18.0 (17.3, 18.6)	0.1115 ^2^
Rural-Urban Continuum Codes, *n* (row %) ^6^			0.4656 ^1^
Metropolitan area—1 million population or more	29 (22.3)	101 (77.7)	
Metropolitan area—250,000 to 1 million population	20 (14.3)	120 (85.7)	
Metropolitan area—fewer than 250,000 population	3 (12.5)	21 (87.5)	
Nonmetropolitan urban area or rural area adjacent to metropolitan area	8 (20.5)	31 (79.5)	
Nonmetropolitan urban area or rural area not adjacent to metropolitan area	3 (15.0)	17 (85.0)	

CL = confidence limits. ^1^ Fisher’s exact test. ^2^ Wilcoxon two-sample test, one-sided *p*-value. ^3^
*n* = 353; 5 individuals with missing county. United States Census Bureau. https://www2.census.gov/programs-surveys/popest/datasets/2010-2018/counties/asrh/cc-est2018-alldata.csv (accessed on 6 June 2021). ^4^
*n* = 353; 5 individuals with missing county. U.S. Department of Agriculture Economic Research Service. U.S. Department of Agriculture county-level datasets. https://www.ers.usda.gov/data-products/county-level-data-sets/ (accessed on 6 June 2021). ^5^
*n* = 353; 5 individuals with missing county. United States Census Bureau. https://www.census.gov/data/datasets/2015/demo/saipe/2015-state-and-county.html (accessed on 6 June 2021). ^6^
*n* = 353; 5 individuals with missing county. USDA Economic Research Service. U.S. Department of Agriculture Rural-Urban Continuum Codes. https://www.ers.usda.gov/data-products/rural-urban-continuum-codes// (accessed on 6 June 2021).

**Table 3 children-08-00835-t003:** Clinical characteristics by any clinical trial participation in the Muscular Dystrophy Surveillance, Tracking, and Research Network (MD STAR*net*), 2000–2015.

Characteristics	In Clinical Trial	Not in Clinical Trial	*p*-Value
Total	64	294	
Ages (years), Mean (95% confidence limits)			
First visit ^1^	5.1 (4.4, 5.7)	4.9 (4.5, 5.2)	0.2586 ^2^
Last visit	9.6 (8.9, 10.3)	9.3 (8.9, 9.7)	0.4015 ^2^
Ambulation status, *n* (row %)			0.0225 ^3^
Not walking	11 (10.6)	93 (89.4)	
Walking	53 (20.9)	201 (79.1)	
Non-trial corticosteroid use, *n* (row %)			<0.0001 ^3^
No	5 (4.8)	100 (95.2)	
Yes	59 (23.3)	194 (76.7)	
Family history, *n* (row %)			0. 3832 ^3^
Definite	18 (14.9)	103 (85.1)	
Suspected	0 (0.0)	8 (100.0)	
No known	8 (22.2)	28 (77.8)	
Unknown	38 (19.7)	155 (80.3)	

^1^ Missing first visit age (*n* = 3). ^2^ Wilcoxon two-sample test, one-sided *p*-value. ^3^ Fisher’s exact test.

## Data Availability

Due to privacy concerns (detailed personal information was obtained from a small number of individuals living in a defined surveillance area), data from the MD STAR*net* is not publicly available. Data used for this analysis are maintained at the Centers for Disease Control and Prevention.

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
