# Peer review of "Characteristics of Clinical Trial Participants with Duchenne Muscular Dystrophy: Data from the Muscular Dystrophy Surveillance, Tracking, and Research Network (MD STARnet)"

_children, 2021, doi:10.3390/children8100835_

Round 1

Reviewer 1 Report

Enjoyable reading 

A very different, yet important, approach in looking at clinical trials accessibility and feasibility in the real world. 

I recommend better data visualization of timelines, and tables (especially those involving geographic areas) into geospatial heat maps with R.

Reviewer 2 Report

The authors have performed a surveillance study on the DMD/BMD patients from the six different sites to determine the predictors of clinical trial participation. The authors have mentioned about the potential limitation of the study in the manuscript. The study is mostly complete and the conclusions drawn are well supported by the result. I have some minor comments which are provided below

  1. The introduction section is very short. Please include a brief description about DMD/BMD and previous studies related to it.
  2. The clinical manifestation of DMD is very broad and the symptoms associated with it overlap with many other muscular dystrophy. Please includes the description about the various symptoms of DMD/BMD and what are the exclusion criteria included to select only the patients of DMD/BMD in this study?
  3. Please include a flow chart of the study.
